# Vitamin D status among adolescents in Kuwait: a cross-sectional study

Abdullah Al-Taiar,[1] Abdur Rahman,[2] Reem Al-Sabah,[1] Lemia Shaban,[2] Anwar Al-Harbi[3]

► Additional material is published online only. To view please visit the journal online (http://dx.doi.org/10.1136/bmjopen-2017-021401).

[1]Department of Community Medicine and Behavioural Sciences, Faculty of Medicine, Kuwait University, Kuwait, Kuwait
[2]Department of Food Science and Nutrition, College of Life Sciences, Kuwait University, Kuwait, Kuwait
[3]Department of Science and Nutrition, Kuwait Institute for Scientific Research, Kuwait, Kuwait

**Correspondence to**
Dr Abdur Rahman;
abdurrahman.ahmad@ku.edu.kw

## ABSTRACT

**Objectives** In Kuwait, as in many Arab states in the Gulf region, there are limited data on the prevalence of vitamin D deficiency among healthy adolescents. This study aimed to estimate the prevalence of vitamin D deficiency in a nationally representative sample of adolescents and investigate factors associated with vitamin D status.

**Methods** A cross-sectional study was conducted on 1416 adolescents aged 11–16 years, who were randomly selected from middle schools in all governorates of Kuwait. Data were collected from parents through self-administered questionnaire and from adolescents through face-to-face interview. Vitamin D was measured using liquid chromatography-tandem mass spectrometry. Logistic regression was used to investigate the independent factors associated with vitamin D status.

**Results** The prevalence of vitamin D deficiency was 81.21% (95% CI 71.61% to 90.81%), while severe deficiency was 39.48%. Only 3.60% of adolescents were vitamin D-sufficient. The prevalence of vitamin D deficiency was significantly higher among girls compared with boys (91.69% vs 70.32%; p<0.001). There was a significant inverse correlation between vitamin D and parathyroid hormone (Spearman correlation=−0.35; p<0.001). In the final model, gender, age, governorate, parental education, body mass index, vitamin D supplement and the number of times adolescents walk to schools per week were all significantly related to vitamin D deficiency.

**Conclusion** High prevalence of vitamin D deficiency was noted among adolescents in Kuwait despite the abundant sunshine, which may reflect strong sun avoidance behaviour. Adequate outdoor daytime activities should be encouraged especially for girls. We call for locally tailored guidelines for vitamin D supplement in which girls should have a higher dose compared with boys.

## INTRODUCTION

An optimal vitamin D status is extremely important during adolescence for proper growth and bone mineral accrual.[1] In addition to its critical role in bone health, several observational studies have linked vitamin D deficiency to various disease conditions during adolescence or adulthood, such as asthma and allergies,[2] type 2 diabetes,[3] depression,[4] cancer and even all-cause mortality.[5] However, the results of randomised control trials which tested the impact of vitamin D supplement

### Strengths and limitations of this study

► This is the first study that aimed to estimate the prevalence of vitamin D on a large nationally representative sample of healthy adolescents in Kuwait.
► We measured vitamin D using the recommended laboratory methodology and gathered data from both parents and adolescents.
► We did not measure skin colour, which can be an important determinant for vitamin D synthesis in the skin.

on the risk of these disease conditions remain inconclusive. For example, there was no clear evidence from randomised controlled trials that vitamin D supplement reduces the risk of cancer,[6] asthma,[7] cardiovascular diseases,[8] depression[9] or overall mortality.[10]

Several studies have evaluated the prevalence of vitamin D deficiency and insufficiency during adolescence, reporting high prevalence of vitamin D deficiency worldwide.[11 12] This was the case even in countries with abundant sunshine, which is the main factor for endogenous vitamin D synthesis. For example, in India the prevalence of vitamin D deficiency among adolescents was reported to be 85%–98%,[13 14] while in Saudi Arabia it was found to be around 96%.[15] In Arab states in the Gulf region or the broader Middle East, there is paucity of data on vitamin D status in paediatric and adolescent populations. The few studies that showed a high prevalence of vitamin D deficiency in adolescents suffered from major methodological weaknesses.[15–17] These include small sample size, lack of proper sampling technique or use of less optimal laboratory method to measure vitamin D. It is also known that factors which influence vitamin D status such as avoiding sun exposure, skin pigmentation, high body mass index (BMI) and dietary factors, particularly the lack of vitamin D supplement and intake of vitamin D-rich foods,[18] are all on the rise in Arab states in the Gulf region. The aims of the present study were therefore

to assess vitamin D status among adolescents living in Kuwait, a country with a plenty of sunshine, and to investigate factors associated with low vitamin D level.

## METHODS

### Study population and study participants

Kuwait is a small country at a latitude of 29.3759° N with a population of 4.2 million. Approximately, 25% of the population is under the age of 19 years. The study population comprised students between 11 and 16 years old in public middle schools from all governorates of Kuwait. A school-based, cross-sectional study was conducted on students from 12 public middle schools, which were selected using a stratified multistage cluster random sampling with a probability proportional to size. The sample allocation in each governorate was based on the relative size of that governorate as judged by the total number of students in the governorate. Students with major chronic disease conditions are registered in each school, and these were excluded from the study. However, students with minor illnesses or self-reported illnesses were included in the study. The type of illness was recorded during data collection and was included in the analysis.

### Data collection

Data were collected (including blood samples) in February, March and April 2016. An informed consent was sent to the parents along with a self-administered questionnaire on parents' level of education (no formal education, primary/intermediate, high school, diploma, university and above), income, type of housing, number of siblings of the index child, passive smoking at household and the number of times per week the index child had a meal prepared outside home during the last 3 months. Trained dedicated personnel carried out face-to-face interviews with the students using a structured questionnaire. The questionnaire was carefully developed after extensive review of the literature and was pilot-tested on 20 students who were not included in the study. It comprised questions on habitual sun exposure during the last 3 months, which was assessed by the core questions developed to measure exposure to sunlight in adolescents as described by Glanz et al.[19] We also collected data on smoking habits, physical activity and dietary intake. Data on physical activity were collected using a questionnaire that was developed based on the Youth Physical Activity Questionnaire in the UK[20] and The Arab Teens Lifestyle Study.[21] The questionnaire was validated among high school students and showed strong correlation with data collected by accelerometers (Spearman correlation=0.92; p<0.001 for total steps count) (N Al Saied, not published). Data on dietary intake of vitamin D were collected from only 200 students using the Food Frequency Questionnaire for calcium and vitamin D intake in adolescents,[22] which has been validated in our setting.[23] Food models or serving containers were used to assist in estimating serving size. Measurements of standing height and body weight of the study subjects were assessed using a digital weight and height scale (Detecto) in a standardised manner.

### Laboratory methods

Five millilitres of venous blood were collected in gel-containing tubes (SST II Advance, BD Vacutainer) from each participant by a trained nurse, and the samples were protected from light. On the same day, the samples were centrifuged at 2000 × g for 15 min and the serum was transferred to Eppendorf tubes and stored at −80°C until analysis. Serum 25-hydroxyvitamin D (25-OH-D) concentration is the best marker of vitamin D status because it is the major circulating form of vitamin D, reflecting both the amount produced in the skin after sun exposure and the amount consumed in foods.[24] It was measured in a College of American Pathologists-accredited laboratory by liquid chromatography-tandem mass spectrometry, which is the recommended method for vitamin D assessment in epidemiological studies.[25] According to the Endocrine Society[26] and the Society for Adolescent Health and Medicine,[27] we used the following cut-off of 25-OH-D to define vitamin D status: vitamin D deficiency <50 nmol/L (20 ng/mL); vitamin D insufficiency 50–75 nmol/L (20–30 ng/mL); and vitamin D sufficiency ≥75 nmol/L (30 ng/mL). Thus, hypovitaminosis D was defined in the presence of 25-OH-D levels <75.0 nmol/L (30 ng/mL). Furthermore, severe vitamin D deficiency was defined as 25-OH-D levels <25.0 nmol/L (10 ng/mL).[28] Serum intact parathyroid hormone (PTH) was measured using the Access Intact PTH chemiluminescent immunoassay with the UniCel DxI 800 Beckman Coulter analyser using a commercial kit (cat #A16972). Similar to other studies, PTH level ≥65.0 ng/L was suggestive of hyperparathyroidism,[29] although there is no consensus on this cut-off point.

### Patient and public involvement

The public were not directly involved in the design of this study. However, schools were involved in approaching the parents to obtain their consents. We also sent the results of the laboratory tests of each participant to the parents through schools using closed envelops to ensure confidentiality.

### Data analysis

Data were double-entered into specifically designed database using EpiData Entry. Data analysis was conducted using Stata V.12. BMI was calculated as weight (kg) divided by height squared (m$^2$). Weight status was categorised into normal, overweight and obese according to the WHO growth charts. $\chi^2$ test was used to test differences in categorical variables. Because vitamin D deficiency was a common outcome and the log-binomial model failed to converge, a modified version of multiple logistic regression that calculates prevalence ratio was used to study the association between each presumed risk factor and vitamin D deficiency (25-OH-D <50 nmol/L). Variables that were

found to be statistically significant at a 15% level of significance were considered in the multivariate analysis. Variables were divided into groups (eg, sociodemographic, sun exposure and so on) and were added sequentially to the model. Statistical significance of variables was evaluated by likelihood ratio test, which compares the model with and without the variable. Goodness of fit of the final model was evaluated by the Hosmer-Lemeshow test. Factors with p<0.05 were deemed to be statistically significant. The analysis above was repeated using hypovitaminosis (25-OH-D<75.0 nmol/L) as the binary outcome. We also conducted multiple linear regression analysis using vitamin D level as a continuous outcome (after log transformation) and reported the results in the text. Because of the complex structure of these survey data, we used survey method, which gives more precise estimates of SEs.

## RESULTS

Of 1583 parents approached, 161 refused to participate (because children, parents or both refused). Another six samples were not sufficient to conduct blood analysis; thus, the analysis below comprised 1416 students. Table 1 shows the sociodemographic characteristics of the study group. The mean (SD) age was 12.48 (0.94) years and 694 (49.01%) were boys.

### Prevalence of vitamin D deficiency

The median (IQR) of 25-OH-D was 29.7 (19.2–44.2) nmol/L. The prevalence of vitamin D deficiency was 81.21% (95% CI 71.61% to 90.81%), while severe deficiency was 39.48%. Only 3.60% of the participants were vitamin D-sufficient. The prevalence of vitamin D deficiency was significantly higher among girls compared with boys (91.69% vs 70.32%; p<0.001). Also, the median (IQR) of 25-OH-D was 39.8 (29.4–52.7) nmol/L and 21.5 (14.7–30.7) nmol/L among boys and girls, respectively (p<0.001). There was no difference in the prevalence of vitamin D deficiency between Kuwaitis and non-Kuwaitis after stratification by gender. There was significant inverse correlation between 25-OH-D and PTH (Spearman correlation=−0.35; p<0.001). In relation to vitamin D status, elevated PTH (secondary hyperparathyroidism PTH ≥65 ng/L, that is, 6.89 pmol/L) was detected in 55.81%, 31.30% and 27.57% of adolescents with vitamin D severe deficiency, deficiency and insufficiency, respectively. If we used the reference range of the laboratory where the samples were tested (PTH: 1.3–9.3 pmol/L), the prevalence of secondary hyperparathyroidism would be 32.56%, 12.35% and 11.68% among adolescents with vitamin D severe deficiency, deficiency and insufficiency, respectively. The median (IQR) PTH was significantly different, dependent on vitamin D status (severe deficiency 7.37 (5.44–10.55) pmol/L; deficiency 5.63 (4.13–7.40) pmol/L; insufficiency 5.14 (3.87–6.99) pmol/L; sufficiency 4.31 (3.75–5.73) pmol/L; p<0.001). The actual relationship between 25-OH-D and PTH levels is shown in figure 1.

### Factors associated with vitamin D status

Factors that showed significant association with vitamin D deficiency (25-OH-D<50 nmol/L) in univariate analysis were gender, parental education, the number of times per week the participants consumed breakfast or dinner prepared outside home and taking supplements, walking to/from school (instead of using school bus or car), in addition to BMI (online supplementary table). Similarly, time spent outdoor per day (between 10:00 and 16:00) during weekdays and weekends, in addition to wearing sunscreen and staying in shade or under an umbrella, were all significant predictors in univariate analysis.

Table 2 shows the independent predictors for vitamin D deficiency in multivariate analysis. In the final model, gender, age, governorate, parental education, vitamin D supplement and the number of times the participant walks to/from school per week were all significantly related to vitamin D deficiency. Data from the Food Frequency Questionnaire for calcium and vitamin D intake were available only for 200 study subjects, and of all food items, only eggs were significantly related to vitamin D deficiency in multivariate model (data not shown). We repeated the analysis above using hypovitaminosis (sufficient vs insufficient/deficient) as a binary outcome. The same factors were found to be significantly related to hypovitaminosis, although mother's education (instead of father's education) was found to be significant in this analysis. We also used linear regression to identify the factors associated with vitamin D level as a continuous variable. Factors that showed significant association with vitamin D level in multivariate analysis were age, gender, governorate, passive smoking at household, number of times per week the participants consumed breakfast prepared outside home in the last 3 months, time spent outdoor during weekdays, preference to stay in shade or under the umbrella, and BMI categories. With respect to dietary factors, milk and cheese, fast food hamburger and tuna fish were significantly associated with vitamin D level in univariate linear regression. When these food items were introduced to the final model, only milk and tuna fish were significantly associated with vitamin D level. It is worth noting that the best model was able to explain only 45% of the variability in vitamin D level in this analysis.

## DISCUSSION

Although several reports have described vitamin D deficiency among adults in the Arab states in the Gulf region, there are limited data on the prevalence of vitamin D deficiency among healthy adolescents.

We used a nationally representative sample to examine vitamin D levels in a large number of healthy adolescents, becoming to our knowledge the largest study focusing on this group in Kuwait. We used the recommended method to measure vitamin D level and demonstrated that vitamin D deficiency is too common despite the abundant sunshine. Secondary hyperparathyroidism due to vitamin

**Table 1** Sociodemographic characteristics and vitamin D levels among adolescents in public middle schools in Kuwait

| Characteristics | Boys, n=694 | | Girls, n=722 | | Total, N=1416 | |
|---|---|---|---|---|---|---|
| Age in years, mean (SD) years | 12.56 | (0.94) | 12.41 | (0.92)* | 12.48 | (0.94) |
| 25-OH-D nmol/L, median (IQR) | 39.80 | (29.4–52.7) | 21.50 | (14.7–30.7)* | 29.70 | (19.2–44.2) |
| | **n** | **(%)** | **n** | **(%)** | **n** | **(%)** |
| Nationality (n=1416) | | | | | | |
| Kuwaiti | 423 | (60.95) | 658 | (91.14)* | 1081 | (76.34) |
| Non-Kuwait | 271 | (39.05) | 64 | (8.86) | 335 | (23.66) |
| Father's education (n=1383) | | | | | | |
| No formal education | 9 | (1.33) | 6 | (0.85)* | 15 | (1.08) |
| Primary/Intermediate | 95 | (14.05) | 126 | (17.82) | 221 | (15.98) |
| Secondary (high school) | 157 | (23.22) | 187 | (26.45) | 344 | (24.87) |
| Diploma | 126 | (18.64) | 135 | (19.09) | 261 | (18.87) |
| University and above | 289 | (42.75) | 253 | (35.79) | 542 | (39.19) |
| Mother's education (n=1396) | | | | | | |
| No formal education | 18 | (2.64) | 13 | (1.82) | 31 | (2.22) |
| Primary/Intermediate | 79 | (11.60) | 73 | (10.21) | 152 | (10.89) |
| Secondary (high school) | 140 | (20.56) | 164 | (22.94) | 304 | (21.78) |
| Diploma | 136 | (19.97) | 168 | (23.50) | 304 | (21.78) |
| University and above | 308 | (45.23) | 297 | (41.54) | 605 | (43.34) |
| Father's income (Kuwaiti dinars) (n=1370) | | | | | | |
| Less than 500 | 64 | (9.61) | 27 | (3.84)* | 91 | (6.64) |
| 500–1000 | 168 | (25.23) | 136 | (19.32) | 304 | (22.19) |
| 1001–1500 | 175 | (26.28) | 246 | (34.94) | 421 | (30.73) |
| 1501–2000 | 100 | (15.02) | 119 | (16.90) | 219 | (15.99) |
| More than 2000 | 83 | (12.46) | 90 | (12.78) | 173 | (12.63) |
| Do not wish to tell | 76 | (11.41) | 86 | (12.22) | 162 | (11.82) |
| Mother's employment status (n=1388) | | | | | | |
| Housewife | 267 | (39.38) | 221 | (31.13)* | 488 | (35.16) |
| Paid employment | 307 | (45.28) | 373 | (52.54) | 680 | (48.99) |
| Others | 104 | (15.34) | 116 | (16.34) | 220 | (15.85) |
| Housing (n=1397) | | | | | | |
| Rented flat | 301 | (44.20) | 209 | (29.19)* | 510 | (36.51) |
| Rented house | 94 | (13.80) | 69 | (9.64) | 163 | (11.67) |
| Owned flat | 17 | (2.50) | 42 | (5.87) | 59 | (4.22) |
| Owned house | 269 | (39.50) | 396 | (55.31) | 665 | (47.60) |

All %s are column percentages.
*P<0.05, that is, significant difference between boys and girls at the 5% level of significance.
25-OH-D, 25-hydroxyvitamin D.

D deficiency is also common and girls had lower vitamin D levels compared with boys.

In our setting, approximately, 81% and 15% of adolescents were vitamin D-deficient or vitamin D-insufficient, respectively. This is higher than that reported among adolescents in European countries,[30–32] Southeastern USA[33 34] or New Zealand.[35] Our finding is more akin to that reported from Saudi Arabia (95.6%),[15] India (85%–98%),[13 14] or Korea where only 2.3% of adolescents were found to be vitamin D-sufficient.[36] Our findings are also

in accordance with the high prevalence reported consistently among adult populations in Kuwait[37] or other countries in the Gulf region such as Saudi Arabia,[38] Bahrain[39] and Qatar.[40]

Similar to other studies among adolescents,[29 33 41] our findings showed that a large number of those with low vitamin D level had evidence of secondary hyperparathyroidism. We reported a much higher prevalence of secondary hyperparathyroidism compared with other studies,[29 41] which could be due to using lower PTH cut-off

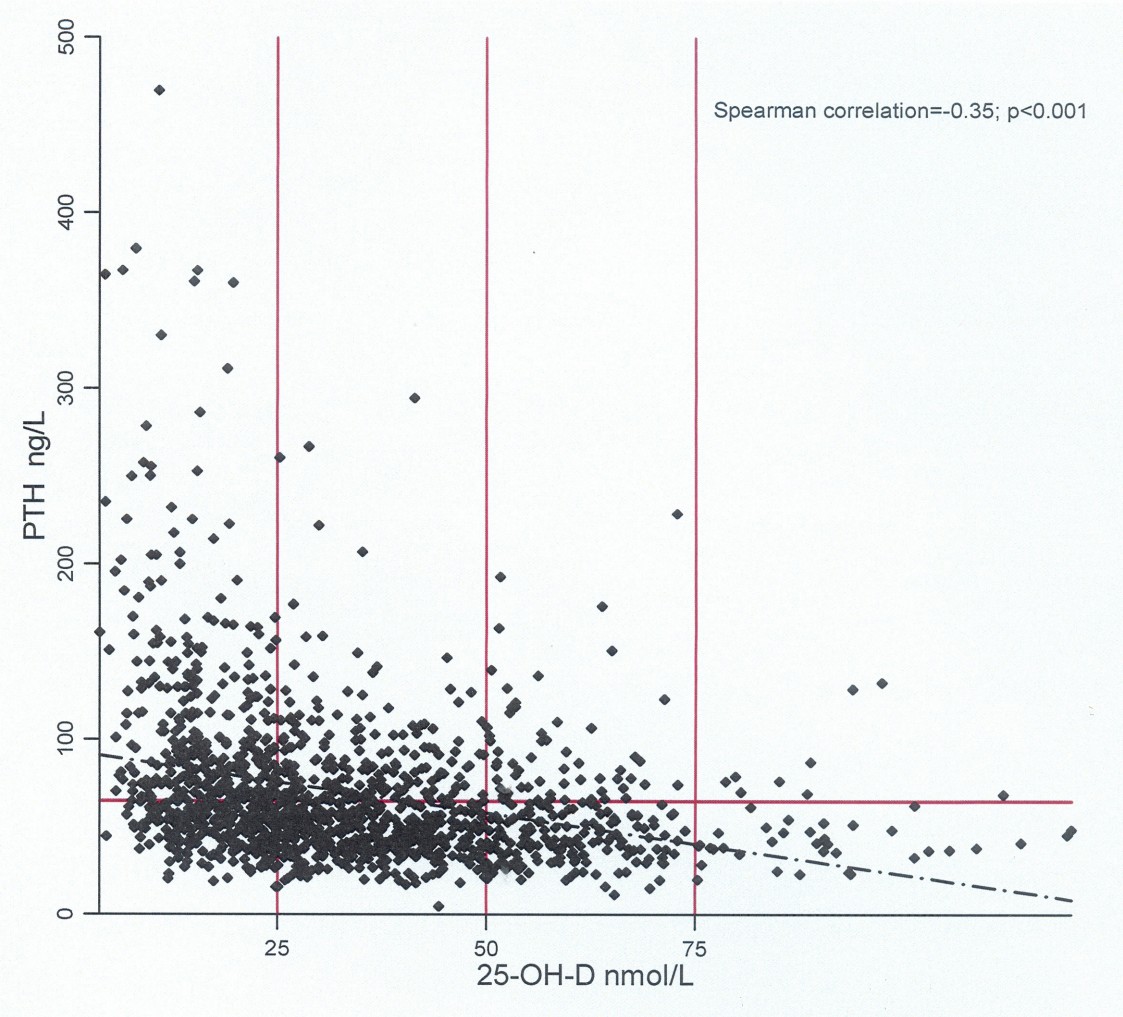

**Figure 1** Parathyroid hormone (PTH) with serum 25-hydroxyvitamin D (25-OH-D) concentration. Individuals above the horizontal line (PTH ≥65.0 ng/L) are adolescents with secondary hyperthyroidism. Vertical lines represent the 25-OH-D cut-off used to define vitamin D deficiency, insufficiency and sufficiency (seven participants with 25-OH-D >150 nmol/L were omitted).

point to define secondary hyperthyroidism or using different laboratory methods to measure PTH. Hyperparathyroidism, which potentially leads to increased bone resorption, is particularly important during adolescence because it is the critical period to achieve peak bone mass. Serum PTH decreased as 25-OH-D level increased with no clear threshold of serum 25-OH-D at which PTH level plateaued, although very few participants had elevated PTH when 25-OH-D level was above 75 nmol/L (figure 1). Although we did not measure markers of bone turnover, this provides an additional evidence that 25-OH-D should be maintained above 75 nmol/L, which is the current cut-off point used to define vitamin D sufficiency in adolescents.[26 27] There were, however, adolescents with low vitamin D concentration without secondary hyperparathyroidism (figure 1). This has been noted in other studies,[33 41] the clinical significance of which is unknown in children and deserves further investigation.

Because of the abundant sunshine in the Arab states in the Gulf region, there has been a lot of scepticism of vitamin D deficiency. Informal discussions, even among

scientific community, in the region show strong denial, attributing such low levels of vitamin D to poorly executed or faulty laboratory measurements. Our findings should eliminate doubts on vitamin D deficiency among adolescents in Kuwait and pave the way to look for practical solutions. Factors that may hinder cutaneous synthesis of vitamin D despite abundant sunshine may include dust storms (which occur almost in one-third of a year in Kuwait[42]), an avoidance of sun exposure by indoor lifestyle, head/body covering or other cultural practices, and the use of sunscreen. Self-reported exposure to sunlight was a significant predictor for vitamin D deficiency in univariate analysis but lost statistical significance in multivariate analysis. Most of the adolescents in our setting avoid sunlight as only around 6% of the participants spent more than 2 hours outdoor per day between 10:00 am and 16:00 during weekdays (data not shown). Avoidance of sunlight has been suggested to be the underlying reason for the counterintuitive seasonal variation that has been reported in the region, where the prevalence of vitamin D deficiency is higher in the summer compared with the

**Table 2** Factors associated with vitamin D deficiency among adolescence in multivariate analysis

| Characteristics | Total | Prevalence of vitamin D deficiency | | Prevalence ratio (95% CI) | | P values |
|---|---|---|---|---|---|---|
| | | n | (%) | | | |
| Gender | | | | | | |
| Male | 694 | 488 | (70.32) | 1 | (Ref) | <0.001 |
| Female | 722 | 662 | (91.69) | 1.20 | (1.17–1.21) | |
| Age (year) | | | | | | |
| <12 | 527 | 409 | (77.61) | 1 | (Ref) | 0.003 |
| 12 | 439 | 369 | (84.05) | 1.10 | (1.04–1.14) | |
| ≥13 | 450 | 372 | (82.67) | 1.07 | (1.01–1.12) | |
| Governorate | | | | | | |
| Capital | 156 | 127 | (81.41) | 1 | (Ref.) | <0.001 |
| Hawally | 246 | 188 | (76.42) | 0.88 | (0.73–1.01) | |
| Farawanya | 236 | 183 | (77.54) | 0.85 | (0.68–0.99) | |
| Jahra | 239 | 214 | (89.54) | 1.10 | (0.98–1.17) | |
| Mubarak Al-Kabeer | 148 | 124 | (83.78) | 1.06 | (0.93–1.14) | |
| Ahmadi | 371 | 296 | (79.78) | 1.06 | (0.95–1.13) | |
| Father's education | | | | | | |
| Primary/intermediate/no formal education | 236 | 205 | (86.86) | 1 | (Ref) | <0.009 |
| Secondary (high school) | 344 | 297 | (86.34) | 1.04 | (0.92–1.12) | |
| Diploma | 261 | 220 | (84.29) | 1.01 | (0.88–1.10) | |
| University and above | 542 | 399 | (73.62) | 0.89 | (0.76–1.01) | |
| Passive smoking in household | | | | | | |
| No | 901 | 704 | (78.14) | 1 | (Ref) | <0.006 |
| Yes | 489 | 426 | (87.12) | 1.08 | (1.02–1.13) | |
| Currently taking supplements | | | | | | |
| No | 1256 | 1045 | (83.20) | 1 | (Ref) | <0.001 |
| Yes | 158 | 103 | (65.19) | 0.63 | (0.49–0.76) | |
| Consumption of sugary drinks per week | | | | | | |
| None | 171 | 123 | (71.93) | 1 | (Ref) | 0.002 |
| 1–3 times | 772 | 633 | (81.99) | 1.11 | (1.05–1.16) | |
| 4–6 times | 171 | 135 | (78.95) | 1.07 | (0.96–1.14) | |
| 7 or more times | 299 | 256 | (85.62) | 1.15 | (1.08–1.19) | |
| Number of times walking to/from school per week | | | | | | |
| None | 1158 | 982 | (84.80) | 1 | (Ref) | <0.001 |
| 1–8 times | 155 | 99 | (63.87) | 0.81 | (0.68–0.93) | |
| Every day | 103 | 69 | (66.99) | 0.77 | (0.61–0.92) | |
| Body mass index categories* | | | | | | |
| Normal weight | 601 | 465 | (77.37) | 1 | (Ref) | 0.001 |
| Overweight | 320 | 270 | (84.38) | 1.08 | (1.01–1.13) | |
| Obese | 471 | 400 | (84.93) | 1.12 | (1.07–1.16) | |
| Underweight | 24 | 15 | (62.50) | 0.93 | (0.67–1.10) | |

%s are row percentages.
*According to the WHO growth charts.
Ref, reference.

winter season.[41] Owing to the desert climate, avoiding sun exposure would be stronger during summer due to extremely high temperature. We collected blood samples in February, March and April; thus, we were unable to describe seasonal variation in vitamin D status. Exposure to sunlight contributes up to 90%–95% of the vitamin D supply, while the number of foods naturally containing a significant quantity of vitamin D is very limited, except for some oily fish that is rarely consumed by adolescents worldwide and in our setting[43] (~5% of adolescents in our setting consumed salmon once per week or more). As such, education of parents and adolescents on the safe amount of sun exposure, in addition to changes in the school environment to facilitate exposure to sunlight, may produce the highest impact on vitamin D status among adolescents. There is no consensus on the duration of time at which adolescents can safely be exposed to direct sunlight. However, exposure of legs and arms for at least 15 min twice per week has been reported to be sufficient for adequate sun-induced cutaneous vitamin D synthesis in adolescents.[43]

Similar to other studies that investigated vitamin D deficiency among adolescents, girls had a significantly higher prevalence of vitamin D deficiency compared with boys even after adjusting for all potential confounders. Also, vitamin D level was significantly lower among girls compared with boys. Similar findings have been reported among adolescents from Saudi Arabia,[44] India,[14] Korea[36] and Taiwan.[45] This pattern is not common in other settings as reported from a clinic-based, cross-sectional study in the USA[33] or Italy.[31] In the Arab states in the Gulf region and India, cultural practices, such as the type of clothing that covers the whole body, and indoor lifestyle have been proposed to exacerbate vitamin D deficiency among girls. Indeed, we found boys consistently reporting higher amount of outdoor exposure to sunlight compared with girls. In addition, girls reported consistently higher sun avoidance behaviours such as using sunscreen and staying in shade or under an umbrella. Adjusting for self-reported exposure to sunlight did not explain the whole association between vitamin D status and gender, which could be due to residual confounding. We also collected data on the type of clothing among girls using photo cards (no head covering, head covering, head and face covering), but vitamin D deficiency was too common among girls, which did not allow for us to investigate the impact of the type of dress on vitamin D status among girls. In our setting, efforts to improve vitamin D level among adolescents should focus on girls through encouragement of safe amount of exposure to sunlight and/or intake of vitamin D-rich foods. Also, the current guidelines for vitamin D supplement should take the difference between girls and boys into account and recommend higher doses for girls compared with boys.

Most of the students in Kuwait do not walk to schools and are mainly transported by cars or school buses. The number of times a student walks to/from school as well as the time spent on walking to/from school were

significantly associated with vitamin D deficiency. Those who walk to/from schools were less likely to have vitamin D deficiency compared with those who used a school bus or car. Walking to/from schools can be a good opportunity for sunlight exposure as well as increasing physical activity, which itself may improve vitamin D level. In a study in the USA, physical activity was inversely associated with hypovitaminosis D but not related to vitamin D level,[33] while in another study vigorous physical activity was significantly associated with vitamin D level.[34] It has been proposed that physical activity may increase the level of vitamin D through increasing the time spent outdoor in sunlight or through reducing the risk of obesity.[18] However, in our study the total time spent on other physical/sport activities was significantly associated with vitamin D deficiency in univariate but not in multivariate analysis. Unlike walking to/from schools, most sport activities in our setting are practised indoor or during night-time due to severe hot weather, which may explain the lack of association between total time spent on physical activities and vitamin D deficiency in multivariate analysis. Like many other studies,[30 33 36] BMI was inversely associated with vitamin D level, which has been attributed to the sequestration of vitamin D within the plentiful adipose tissue.[46 47] It has also been suggested that leptin, an adipocyte-derived hormone, might activate a pathway that inhibits renal synthesis of the active form of vitamin D.[48]

In our study, sugary drinks were positively associated with the prevalence of vitamin D deficiency. This is similar to that reported from Saudi Arabia[41] and USA.[33] Consumption of sugary drinks among adolescents usually occurs at the expense of milk consumption, which contains vitamin D.[49] As a result, it has been proposed that the availability of vitamin D-fortified juices may help reduce vitamin D deficiency in adolescents.[33] In our data those who consumed sugary drinks were less likely to consume milk (data not shown). Although milk consumption was not related to vitamin D deficiency, it was significantly related to the vitamin D level in linear regression analysis.

We found an association between passive smoking in the household (reported by parents) and vitamin D deficiency among adolescents, which is interesting and has been reported before.[45] Only less than 1% of the adolescents in our setting reported active smoking. Smoking has also been reported to be a significant determinant for low vitamin D levels among adults in some studies[32] but not in others,[50] and this issue remains controversial.[32] Furthermore, a recent study found no significant association between cotinine level and vitamin D deficiency among Korean adolescents.[36] One of the hypothesised mechanism is that smoking may reflect an overall less healthy lifestyle, including less physical activity and poor dietary habits.[32] A causal link has also been proposed as metabolites in cigarette's smoke can inhibit CYP27A1 activity,[51] which is involved in vitamin D metabolism.

## Strengths and limitations

This is the first study that measured vitamin D in a large nationally representative sample of adolescents in Kuwait. We used a measurement method that is recommended in epidemiological studies and gathered data from both parents and adolescents. However, we did not measure skin colour, which can be an important determinant for vitamin D synthesis in the skin during exposure to sunlight. We evaluated the dietary intake of vitamin D in only a subgroup of study participants. However, dietary intake of vitamin D is known to have a smaller contribution to vitamin D status compared with cutaneous synthesis in response to sun exposure,[43] and data on vitamin D supplement were obtained from all participants.

## CONCLUSION

In conclusion, a high prevalence of vitamin D deficiency was noted among adolescents in Kuwait despite the abundant sunshine. This may reflect strong sun avoidance behaviour as only less than 4% of adolescents were vitamin D-sufficient and most of them were on vitamin D supplements. An optimal vitamin D level is essential during adolescence, and vitamin D deficiency at this age represents a public health problem that should be addressed. First, our findings should help clear doubts on vitamin D deficiency among adolescents in Kuwait, which is a prerequisite for prevention initiatives. Second, sun exposure is the main source for vitamin D in this age group; therefore, adequate outdoor daytime activities should be encouraged especially for girls. Slight modifications in school environment to facilitate exposure to sunlight during school breaks between classes can be fruitful. With respect to vitamin D supplementation, we call for locally tailored guidelines in which girls should have a higher amount of vitamin D supplement compared with boys. Vitamin D fortification particularly for food products that are popular for adolescents in our setting should also be considered. Increasing exposure to sunlight and food fortification with vitamin D should not be considered mutually exclusive and both strategies can be adopted.

**Acknowledgements**  The authors acknowledge the assistance of Nadien Sameeh Rushdi and the team of data collectors. We also acknowledge the cooperation of all participating schools and the facilitation of the project by the Ministry of Education. We also appreciate the support and cooperation of the staff and management of the United Genetics Laboratories (Kuwait) during vitamin D analysis.

**Contributors**  AA-T contributed to the study design and data collection, analysed the data and drafted the paper. He had full access to all the data and took responsibility for the integrity of the data. AR contributed to the design of the study and data collection, in addition to writing and revising the manuscript. RA-S and LS contributed to the design of the study and data collection, in addition to revising the manuscript with significant intellectual input. A-AH contributed to the data collection and revised the manuscript.

**Funding**  The work was supported and funded by Kuwait University Research Project No WF 02/13.

**Competing interests**  None declared.

**Patient consent**  Obtained.

**Ethics approval**  The study was approved by the Ethics Committee at the Ministry of Health in Kuwait (ref no 2015/248) and the Ethics Committee at Health Sciences Centre, Kuwait University (ref no DR/EC/2338).

**Provenance and peer review**  Not commissioned; externally peer reviewed.

**Data sharing statement**  No additional data are available.

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
