## [Reviewer comments · BMJ Open]

ARTICLE DETAILS

TITLE (PROVISIONAL)	Vitamin D Status among Adolescents in Kuwait: A Cross-sectional Study.
AUTHORS	Al-Taiar, Abdullah; Rahman, Abdur; Al-Sabah, Reem; Shaban, Lemia; Al-Harbi, Anwar

VERSION 1 – REVIEW

REVIEWER	Kelly Virecoulon Giudici Postdoctoral researcher at the Nutrition Department, School of Public Health, University of São Paulo, Brazil
REVIEW RETURNED	15-Jan-2018

GENERAL COMMENTS	The objectives of the study do not bring very innovative information on vitamin D to general scientific community, but contribute to evaluate vitamin D status of young people from a specific region of the world, where studies analyzing vitamin D among adolescents are still less frequent. In addition, the high prevalence of vitamin D among the studied sample highlights the discussion on why even very sunny countries deal with this public health issue. The manuscript would benefit from a general English revision. In addition, inconsistencies found in the results (text and tables) make it important to carefully revise every information. Introduction • Authors say: “However, the impact of vitamin D supplement on the risk of these diseases has not yet been tested in randomized control trials and thus remain under intense debate.” – Many randomized control trials evaluating vitamin D supplementation and health outcomes have already been done with adults, and some others with adolescents. Please update your Introduction accordingly.• “The few studies that showed a high prevalence of vitamin D deficiency suffered from major methodological weaknesses.” – Which studies are those? Please explain what the major methodological weakness are.• The research question or study objective is clearly defined in the abstract, but not in the manuscript text. In the abstract, authors say that the objectives include investigating factors associated with vitamin D status. However, this is not mentioned in the objectives (at the end of Introduction). Methods • In the first paragraph of Discussion, authors say: “Our study used a nationally representative sample to examine a large number of healthy adolescents”. However, the Methods section does not inform if any chronic disease (besides obesity) was considered as an exclusion criterion.• “Measurements of standing height and body weight of the study
---

subjects were assessed using digital weight and height scale in a standardized manner.” – Please provide the manufacturer’s information for the equipment.

- “ (...) the samples were centrifuged at 2000 x g for 15 minutes” – would it be 2000 rpm (rotations per minute)?
- “PTH level \geq 65.0 ng/L was suggestive of hyperparathyroidism.” – Please give the reference.

Results

- In Table 1, I suggest including two columns (male and female), in addition to the total column. It would also be valuable to include 25(OH)D values (in all columns: total, male and female).
- In the text, authors say “694 (49.01%) were females”. However, in Table 1 we can see the information that 694 were male. Please double check all numerical results and correct where necessary.
- In the text: “Prevalence of vitamin D deficiency was 1,150 (81.21%, 95%CI: 71.61-90.81) while severe deficiency was 559 (39.48%). Only 51 (3.60%) were vitamin D sufficient. Prevalence was significantly higher among females compared to males (91.69% vs. 70.32%; $p < 0.001$).” – Which prevalence does the second sentence refers to: vitamin D deficiency, severe deficiency or sufficiency? Analyzing the percentage, one can presume that it refers to vitamin D deficiency. However, the sentence is not clearly written.
- Please rewrite the following sentence to avoid redundancy: “Factors that showed significant association with vitamin D deficiency (25-OH-D $<$ 50 nmol/L) in univariate analysis were Gender, parental education, number of times per week the participants consumed breakfast and dinner prepared outside home, taking supplements, walking to school (instead of using school bus or car) in addition to BMI were all significant predictors in univariate analysis (Supplementary Table).”
- When describing the findings about significant independent predictors for vitamin D deficiency in multivariate analysis, authors say: “Dietary intake data were available only for a few hindered, and of all food items, only eggs were significantly related to vitamin D deficiency in multivariate model.” However, this information is not given in Table 2. On the other hand, other dietary information is given in table 2 but not mentioned in the text (consumption of sugary drinks per week, $p < 0.002$). The term “hindered” is not appropriate. Please replace accordingly.
- Tables 1, 2 and supplementary table: Instead of stating the number of missings in the footnote (“1Missing for 33 participants; 2 Missing for 20 participants; 3 Missing for 46 participants; 4Missing for 28 participants; 5Missing for 19 participants”), I suggest including the real “n” for each variable with parenthesis right after the variable’s name Ex: “Father’s education (n = 1383)”.
- Also, the number of missings seem to be incorrect in some variables. Example: in table 2, title says “1,416 middle school students”. However, there are 488 boys and 662 girls (488 + 662 = 1,150) and no missing information is given for gender. For “Governorate”, the sum is 127+188+183+214+124+296=1,132. Adding the number of missings (1,132 + 20), we get 1,552, and not 1,416 as said in the title. This fact also happens with other variables, so please revise all numerical information and provide the correct information in all tables.
- The percentages described for all variables in Table 2 and supplementary table 1 are visibly incorrect. Example for BMI categories: if 465 subjects are normal weight, 270 are overweight, 400 are obese and 15 are underweight, the percentages should be,

respectively, 40.4%, 23.5%, 34.8% and 1.3% (summing 100%), and not 77.37%, 84.38%, 84.93% and 62.50% as given. Again, the total n for this variable ($n=465+270+400+15=1,150$), considering additional 26 missings ($1,150+26=1,176$), is different from the title ($n=1,416$).

- Figure 1 should present the correlation line and the correlation coefficient between vitamin D and PTH. In addition, please add the unit of measure for 25(OH)D (horizontal axis).

- In supplementary Table 1, please inform the p-values for “Wearing sunscreen” and “Staying in shade or under an umbrella”. If they are not significant, authors may write “NS” and add “NS = not significant” at the footnote.

Discussion

- About the sentence: “Only around 6% of the adolescents spent more than two hours outdoor per day between 10:00 am and 4:00 pm during weekdays (data not shown).” – This data is shown at Supplementary table 1.

- “In our setting, efforts to improve vitamin D level among adolescents should focus on females and the current recommended doses for vitamin D supplement should also take this into account and recommend higher doses for females compared to males.” – Indicating vitamin D supplementation (as any other nutrient supplementation) should involve caution. The best way to promote health is to achieve nutrients recommendation by habitual diet (food intake). For vitamin D, particularities of its metabolism (especially the skin production after sun exposure) gives a slightly different panorama. However, the fact that not many food sources naturally contains good amounts of vitamin D, and that dietary intake use to be insufficient, are not reasons enough to simply recommend supplementation to all ages and health conditions. I believe the discussion of this study should explore more deeply other possible ways to achieve a better vitamin D status among adolescents, and consider supplementation difficulties: it is expensive; adolescents are not usually used to take supplements regularly (low adherence); very high doses may offer risks to health; there are metabolic difference between food / sun made vitamin D and vitamin from supplements, etc.

- Adiposity is known to be a major risk factor to vitamin D deficiency, mainly because this fat soluble vitamin gets stored inside the adipocytes. In your sample, we can find a total of 400 obese adolescents (34.8%) and 270 who were overweight (23.5%). Thus, weight excess corresponds to 58.3% of participants. However, this condition was not discussed in the manuscript. There are only few words remembering that “It has been proposed that physical activity may increase the level of vitamin D (...) through reducing the risk of obesity [11]” and “Like many other studies [22, 25, 28], BMI was inversely associated with vitamin D deficiency.” I believe this theme could be further explored.

- “Like many other studies [22, 25, 28], BMI was inversely associated with vitamin D deficiency.” – This sentence gives an incorrect relationship between BMI and vitamin D status (it says that higher the BMI, less prevalent would be vitamin D deficiency). I believe authors meant to say that BMI was inversely associated with vitamin D concentrations (not with vitamin D deficiency), or in another format, BMI was positively associated with vitamin D deficiency. Please correct the sentence.

- Given that the authors measure PTH and define hyperparathyroidism, I suggest briefly including the physiologic relationship between vitamin D and PTH in the Introduction. Also,

	nothing is discussed about it (it only appears in the “strengths and limitations” section). Thus, it would be interesting to have this topic discussed.  ● A limitation of the study is that “We evaluated the dietary intake of vitamin D in only a subgroup of study participants.” However, this information is not given in the Methods section, and it should be (How many participants had their dietary information collected? Why?). Conclusion  ● Authors discuss about “skepticism about the problem or its clinical significance may hinder efforts to put forward practical solutions. This cynicism particularly among scientific community and decision makers should be cleared.” I believe this skepticism does not come entirely from the idea that vitamin D deficiency is not believed to happen in sunny countries (something like if the measurements were untrue), but the point is that there is a possibility that the actual used reference cutoffs may be overclassifying vitamin D deficiency. Vitamin D reference values were originally based on bone outcomes. Later, vitamin D turned out to be a much more explored hormone, known to act in many other tissues and to participate on a plenty of metabolic pathways. This topic could be better addressed in the Discussion. ● In the sentence: “An optimal vitamin D level is essential during adolescents” – Please replace “adolescents” by “adolescence”. STROBE checklist  ● Items #3, 4, 8, 9, 10, 12C and 16A were not filled in the checklist.
--	--

REVIEWER	Ourania Kolokotroni University of Nicosia Medical School, Nicosia, Cyprus.
REVIEW RETURNED	29-Jan-2018

GENERAL COMMENTS	Paper title: Vitamin D status amongst adolescents in Kuwait: a cross-sectional study BMC Open Comments to the authors General Comments This is a descriptive study that investigated prevalence and determinants of vitamin D deficiency in adolescents in Kuwait. Whilst vitamin D status and predictors of deficiency are well established in some areas of the world and in the adult population, there is still need for more studies on this topic in the adolescent group and in areas of the world such as the Middle East. Adolescence is a critical period for growth and development affected also by changes in behavior that might affect status of vitamin D. Thus this paper extends beyond local interest. The paper would benefit from a linguistic review. Title: Relevant and includes study design. Abstract: Important information is missing from the abstract especially in relation to the methods of the study. For example in the methods section, the authors do not include essential information on participants (e.g. characteristics such as age spectrum, participation rate) and on outcome variables i.e. what data was collected from parents and children and what tools were used to do that? In the results section prevalence should be reported as a percentage and the number of participants in the brackets. In the conclusion, do not include statements such as “which may reflect sun avoidance” unless can support this from your findings.
---

	Strengths and Limitations This section should not include results thus second bullet point is not relevant. Strengths of the study is its study design since cross-sectional studies are appropriate to measure prevalence. Was the participation rate high? If yes, this is also a strength. Introduction: This section is very brief. Whilst the first paragraph sets the scene as to the importance of investigating vitamin D status in adolescents, the rest of this section fails to describe what is already known and what this study will add. For example the authors claim that other studies that investigated vitamin D status (especially in the Middle East area) suffer from methodological problems. What are these problems and how has this current study dealt with these problems in order to provide further evidence on the topic of interest? Also the authors failed to explain how this study might be of international interest. Methodology: The methods section is informative and includes most of the items as per STROBE. Statistical analysis is appropriate and valid. However it will benefit from sub-sections to make it more structured. Information not included but important is:  1. Overall number of adolescents in the age group 11-16 in Kuwait and number of adolescents in the selected sample size i.e. there are x number of adolescents aged 11-16 in all governorates of Kuwait, the overall sample of adolescents attending the 12 selected middle public schools were 1583. 2. Definition and measurement of variables: the authors provide information on how some variables were measured and their definition as per tools used in other studies. For the rest of the variables definition can be found in the tables. It would be good to include in the body text under methods (e.g. education was measured as no education- primary/intermediate –secondary – diploma- university). 3. Season/period of measurements – seasonality is an important determinant of vitamin D status and information on the period of measurement is found in later sections of the paper and not in methods 4. Setting/location: This is not clear and does not allow replication of study methods 5. Reference to patient involvement in the design of the study is not made (as per the journal guidelines) Results: The results section provides a good description of the study results including information on participants (numbers and characteristics) and outcome data reported as population prevalence per vitamin D category (as described in methods). Also authors refer to subsidiary analysis very effectively showing that their data analysis was comprehensive. Tables and figures are clear. Sub sections would provide better structure to this section. Also in regards to this section please note the following:  1. First paragraph: apart from gender and mean age please include other characteristics of participants e.g. ethnicity, parental education and household income) 2. How can the median vitamin D level be 29 nmol/L but 80% of participants have a vitamin D level < 20 nmol/L. How are vitamin D levels distributed in the participants? It might be good inserting a figure that shows the distribution of vitamin D levels amongst participants as a continuous variable. 3. As indicated in the abstract, please report prevalence as % and include (n) in the brackets Discussion:
--	--

	This section requires more structure and a more focused interpretation of the results of the study. Once more sub sections will help. The first paragraph should include a summary of results. The authors attempted to start this section with a summary of results but only referred to vitamin D status and not determinants. Discussion on how vitamin d status compared with other countries is satisfactory. Discussion in regards to the determinants of vitamin D deficiency in Kuwait is discussed next. Generally authors have discussed and interpreted those findings in the context of other studies and the strengths and limitations of their own study. However the discussion on sun exposure is not structured well. Authors start to discuss the skepticism amongst Arab states in regards to the validity of the results which is not part of the aims of the study. They also start to discuss determinants of low sun exposure referring to variables they have not measured such as dust storms and skin type. They need to emphasize first which of the parameters they have measured show that exposure to sun is low and whether these parameters are linked to vitamin D deficiency in their own population (e.g. time spent out in the sun, times walking to school etc.). Authors can then refer to other parameters relating to sun exposure that they have not measured (e.g. skin type, dust storms) and how this might have also contributed to low vitamin D. Conclusion: this section should provide recommendations for the future based on the results of the study. Reference to skepticism and cynicism is not relevant. Instead authors should propose ways on how policy makers can use these results to improve vitamin D status in adolescents in Kuwait.
--	---

VERSION 1 – AUTHOR RESPONSE

Reviewer(s)' Comments to Author:

Reviewer: 1

The objectives of the study do not bring very innovative information on vitamin D to general scientific community, but contribute to evaluate vitamin D status of young people from a specific region of the world, where studies analyzing vitamin D among adolescents are still less frequent. In addition, the high prevalence of vitamin D among the studied sample highlights the discussion on why even very sunny countries deal with this public health issue.

Thank you.

The manuscript would benefit from a general English revision. In addition, inconsistencies found in the results (text and tables) make it important to carefully revise every information.

The manuscript has been now edited and we have clarified inconsistencies. Please see below.

Introduction

- Authors say: “However, the impact of vitamin D supplement on the risk of these diseases has not yet been tested in randomized control trials and thus remain under intense debate.” – Many randomized control trials evaluating vitamin D supplementation and health outcomes have already been done with adults, and some others with adolescents. Please update your Introduction accordingly.

We agree with the comment. We have modified the sentence now saying that Randomized Control Trials have been conducted but their findings remain inconclusive. (Please see introduction, first paragraph, page 5)

- “The few studies that showed a high prevalence of vitamin D deficiency suffered from major methodological weaknesses.” – Which studies are those? Please explain what the major methodological weakness are.

We have cited these studies with their weaknesses now. One of these studies was actually very small, hospital-based, included several nationalities and used less optimal laboratory method to measure vitamin D [1]. The other study used less optimal laboratory method and was not focused on Vitamin D [2]. The third study used improper sampling technique and less optimal laboratory methods [3]. (Please see the last paragraph in The Introduction, page 5)

References:

1. Mansour MM, Alhadidi KM. Vitamin D deficiency in children living in Jeddah, Saudi Arabia. *Indian journal of endocrinology and metabolism* 2012;16:263-9.
2. AlBuhairan FS, Tamim H, Al Dubayee M, et al. Time for an Adolescent Health Surveillance System in Saudi Arabia: Findings From "Jeeluna". *The Journal of adolescent health : official publication of the Society for Adolescent Medicine* 2015;57:263-9.
3. Kaddam IM, Al-Shaikh AM, Abaalkhail BA, et al. Prevalence of vitamin D deficiency and its associated factors in three regions of Saudi Arabia. *Saudi medical journal* 2017;38:381-90.

- The research question or study objective is clearly defined in the abstract, but not in the manuscript text. In the abstract, authors say that the objectives include investigating factors associated with vitamin D status. However, this is not mentioned in the objectives (at the end of Introduction).

We have modified this now. (Please see the end of The Introduction, page 5)

Methods

- In the first paragraph of Discussion, authors say: “Our study used a nationally representative sample to examine a large number of healthy adolescents”. However, the Methods section does not inform if any chronic disease (besides obesity) was considered as an exclusion criterion.

Students with major chronic illness (e.g. congenital heart disease, sickle cell anemia, etc) are registered with the social workers in each school (called special cases). These cases were excluded. However, those with minor illness or self-reported illness were not excluded. We collected data on illness and included this variable in the analysis. (We have added this information to the manuscript, see first paragraph in the methods section, page 6).

There was no significant difference in vitamin D level between those with self-reported illness and those without, [median (IQR) 30.5 (19.2-46.15)] vs [median (IQR) 29.5 (19.3-44.1)], respectively ($p=0.629$). It was thought that excluding the cases of mild illness may introduce bias in assessing the prevalence of vitamin D. As an example, suppose that we excluded students with asthma or allergies- and because asthma/allergies might be related to vitamin D, this is likely to underestimate the prevalence of vitamin D.

- “Measurements of standing height and body weight of the study subjects were assessed using digital weight and height scale in a standardized manner.” – Please provide the manufacturer’s information for the equipment.

We have added this to the method section now (Page 7)

- “ (...) the samples were centrifuged at 2000 x g for 15 minutes” – would it be 2000 rpm (rotations per minute)?

The “g” represent the centrifugal force at which the material is centrifuged. "g" is a function of the speed and the radius of the centrifuge. As the radii of the centrifuges change from laboratory to another, centrifugation speed is usually shown in "g"s rather than rpm in order to standardize the force.

- “PTH level \geq 65.0 ng/L was suggestive of hyperparathyroidism.” – Please give the reference.

We have now written how we came up with this cutoff point. See The Methods section, page 8.

Results

- In Table 1, I suggest including two columns (male and female), in addition to the total column. It would also be valuable to include 25(OH)D values (in all columns: total, male and female).

Table 1 meant to present socio-demographic data of the study group. Including vitamin D level in the table will make the title of table inappropriate (i.e. vitamin D is not a socio-demographic characteristic). The information requested in the comment above are all available in the text. This include vitamin D level among males, females and in the total study group.

- In the text, authors say “694 (49.01%) were females”. However, in Table 1 we can see the information that 694 were male. Please double check all numerical results and correct where necessary.

Thank you for the comment. We corrected the text (page 9).

- In the text: “Prevalence of vitamin D deficiency was 1,150 (81.21%, 95%CI: 71.61-90.81) while severe deficiency was 559 (39.48%). Only 51 (3.60%) were vitamin D sufficient. Prevalence was significantly higher among females compared to males (91.69% vs. 70.32%; $p < 0.001$).” – Which prevalence does the second sentence refers to: vitamin D deficiency, severe deficiency or sufficiency? Analyzing the percentage, one can presume that it refers to vitamin D deficiency. However, the sentence is not clearly written.

We have modified this to become clearer. I.e. vitamin D deficiency not severe deficiency (see first paragraph, page 10)

- Please rewrite the following sentence to avoid redundancy: “Factors that showed significant association with vitamin D deficiency (25-OH-D $<$ 50 nmol/L) in univariate analysis were Gender, parental education, number of times per week the participants consumed breakfast and dinner prepared outside home, taking supplements, walking to school (instead of using school bus or car) in addition to BMI were all significant predictors in univariate analysis (Supplementary Table).”

It seems that you mean the data is presented in the text and the supplementary table. As such, we have shorten this paragraph to one sentence referring to the supplementary table. Because we cannot keep a paragraph with only one sentence, we merged this sentence with the next paragraph. (see The Results section, first paragraph, page 12)

- When describing the findings about significant independent predictors for vitamin D deficiency in multivariate analysis, authors say: “Dietary intake data were available only for a few hindered, and of

all food items, only eggs were significantly related to vitamin D deficiency in multivariate model.” However, this information is not given in Table 2. On the other hand, other dietary information is given in table 2 but not mentioned in the text (consumption of sugary drinks per week, $p < 0.002$). The term “hindered” is not appropriate. Please replace accordingly.

What we meant by dietary intake is the food frequency questionnaire for calcium and vitamin D intake in adolescents as in the reference [4] (please see The Methods section). Consumption of sugary drink is not part of this questionnaire and was available for all study subjects. This questionnaire was conducted only on 200 participants therefore it was not included in the final model because it will reduce the power of the study significantly. When we added the data from food frequency questionnaire to the final model, eggs was the only item significant in the model. We have now stated that food frequency questionnaire for calcium and vitamin D intake was done only for 200 students (see The Methods section, first paragraph, page 7). We also clarified that data from food frequency questionnaire for calcium and vitamin D intake was available for only 200 participants (see Result section, First paragraph, page 12)

Reference:

4. Taylor C, Lamparello B, Kruczek K, et al. Validation of a food frequency questionnaire for determining calcium and vitamin D intake by adolescent girls with anorexia nervosa. *Journal of the American Dietetic Association* 2009;109:479-85, 85 e1-3.

- Tables 1, 2 and supplementary table: Instead of stating the number of missings in the footnote (“1Missing for 33 participants; 2 Missing for 20 participants; 3 Missing for 46 participants; 4Missing for 28 participants; 5Missing for 19 participants”), I suggest including the real “n” for each variable with parenthesis right after the variable’s name Ex: “Father’s education (n = 1383)”.

It would be much easier for readers to assess the impact of missing values if they are mentioned this way. As you can see the number of missing are relatively small compared to our sample and thus the message is that missing are unlikely to change our findings significantly. Presenting real numbers instead of the number of missing will imply that readers have to calculate the number of missing values themselves.

- Also, the number of missings seem to be incorrect in some variables. Example: in table 2, title says “1,416 middle school students”. However, there are 488 boys and 662 girls ($488 + 662 = 1,150$) and no missing information is given for gender. For “Governorate”, the sum is $127+188+183+214+124+296=1,132$. Adding the number of missings ($1,132 + 20$), we get 1,552, and not 1,416 as said in the title. This fact also happens with other variables, so please revise all numerical information and provide the correct information in all tables.

There are no mistakes here. The title of the column where these numbers existed is “prevalence n (%)” hence the next column is prevalence ratio. Because this is a cross-sectional study, the typical data presentation is to show the prevalence of the outcome among each exposure category. There are 488 boys with vitamin D deficiency out of the total number of boys which is 694. Similarly, there were 662 girls with vitamin D deficiency out of 722 girls. In order to remove any confusion, we made the title of the column clearer now saying that this is the “prevalence of vitamin D deficiency”. (See supplementary table and Table 2)

- The percentages described for all variables in Table 2 and supplementary table 1 are visibly incorrect. Example for BMI categories: if 465 subjects are normal weight, 270 are overweight, 400 are obese and 15 are underweight, the percentages should be, respectively, 40.4%, 23.5%, 34.8% and 1.3% (summing 100%), and not 77.37%, 84.38%, 84.93% and 62.50% as given. Again, the total n for this variable ($n=465+270+400+15=1,150$), considering additional 26 missings ($1,150+26=1,176$), is

different from the title (n=1,416).

Please see the response to the comment above. These are not the numbers and %s of participants who are normal weight, overweight or obese. These are the numbers and %s of participants who had vitamin D deficiency in normal weight, overweight and obese. i.e. prevalence of the outcome in each exposure category. Tables can only be understood properly if the column titles and row titles are taken into account.

- Figure 1 should present the correlation line and the correlation coefficient between vitamin D and PTH. In addition, please add the unit of measure for 25(OH)D (horizontal axis).

We have added the regression line to the figure now. We have added the unit to the X-axis. (See figure 1)

- In supplementary Table 1, please inform the p-values for “Wearing sunscreen” and “Staying in shade or under an umbrella”. If they are not significant, authors may write “NS” and add “NS = not significant” at the footnote.

Thank you for the comment. We added this now. (See supplementary Table)

Discussion

- About the sentence: “Only around 6% of the adolescents spent more than two hours outdoor per day between 10:00 am and 4:00 pm during weekdays (data not shown).” – This data is shown at Supplementary table 1.

Indeed this is not shown in the Supplementary table. What is shown in that table is the number of those who had vitamin D deficiency among those who spent more than two hours outdoor per day during weekdays. I.e. Prevalence of vitamin D deficiency in each exposure category. As mentioned above we have made the title of the column very clear now (Prevalence of vitamin D deficiency)

- “In our setting, efforts to improve vitamin D level among adolescents should focus on females and the current recommended doses for vitamin D supplement should also take this into account and recommend higher doses for females compared to males.” – Indicating vitamin D supplementation (as any other nutrient supplementation) should involve caution. The best way to promote health is to achieve nutrients recommendation by habitual diet (food intake). For vitamin D, particularities of its metabolism (especially the skin production after sun exposure) gives a slightly different panorama. However, the fact that not many food sources naturally contains good amounts of vitamin D, and that dietary intake use to be insufficient, are not reasons enough to simply recommend supplementation to all ages and health conditions. I believe the discussion of this study should explore more deeply other possible ways to achieve a better vitamin D status among adolescents, and consider supplementation difficulties: it is expensive; adolescents are not usually used to take supplements regularly (low adherence); very high doses may offer risks to health; there are metabolic difference between food / sun made vitamin D and vitamin from supplements, etc.

The main recommendation that we iterated throughout the manuscript is the exposure to sunlight. As shown in the fourth paragraph in the discussion, where we suggested exposure to sunlight to produce the highest impact “education of the parents on the safe amount of sun exposure in addition to changes in the school environment to facilitate exposure to sunlight may produce the highest impact on vitamin D status among adolescents”. This was the main recommendation in the conclusion and in the abstract. Now, we modified the sentence you mentioned above to highlight this point again. (See The Discussion, Paragraph 5, Page 15)

- Adiposity is known to be a major risk factor to vitamin D deficiency, mainly because this fat soluble vitamin gets stored inside the adipocytes. In your sample, we can find a total of 400 obese adolescents (34.8%) and 270 who were overweight (23.5%). Thus, weight excess corresponds to 58.3% of participants. However, this condition was not discussed in the manuscript. There are only few words remembering that “It has been proposed that physical activity may increase the level of vitamin D (...) through reducing the risk of obesity [11]” and “Like many other studies [22, 25, 28], BMI was inversely associated with vitamin D deficiency.” I believe this theme could be further explored.

The link between obesity and vitamin D has been consistently reported with explanation that you kindly outlined above. Now, we have added the plausible mechanism of this association citing the relevant literature (see The Discussion, page 16)

- “Like many other studies [22, 25, 28], BMI was inversely associated with vitamin D deficiency.” – This sentence gives an incorrect relationship between BMI and vitamin D status (it says that higher the BMI, less prevalent would be vitamin D deficiency). I believe authors meant to say that BMI was inversely associated with vitamin D concentrations (not with vitamin D deficiency), or in another format, BMI was positively associated with vitamin D deficiency. Please correct the sentence.

We agree with the comment. This is corrected now.

- Given that the authors measure PTH and define hyperparathyroidism, I suggest briefly including the physiologic relationship between vitamin D and PTH in the Introduction. Also, nothing is discussed about it (it only appears in the “strengths and limitations” section). Thus, it would be interesting to have this topic discussed.

We agree with the comment and added full paragraph to discuss the relationship between PTH and vitamin D level. (See The Discussion, third paragraph, page 13)

- A limitation of the study is that “We evaluated the dietary intake of vitamin D in only a subgroup of study participants.” However, this information is not given in the Methods section, and it should be (How many participants had their dietary information collected? Why?).

Please see the response to the previous comment. What we meant by dietary intake is the food frequency questionnaire for calcium and vitamin D intake in adolescents. This questionnaire was conducted only on 200 participants. We have now stated that food frequency questionnaire for calcium and vitamin D intake was done only for 200 students (see The Methods section, first paragraph, page 7 & Result section, First paragraph, page 12).

Conclusion

- Authors discuss about “skepticism about the problem or its clinical significance may hinder efforts to put forward practical solutions. This cynicism particularly among scientific community and decision makers should be cleared.” I believe this skepticism does not come entirely from the idea that vitamin D deficiency is not believed to happen in sunny countries (something like if the measurements were untrue), but the point is that there is a possibility that the actual used reference cutoffs may be overclassifying vitamin D deficiency. Vitamin D reference values were originally based on bone outcomes. Later, vitamin D turned out to be a much more explored hormone, known to act in many other tissues and to participate on a plenty of metabolic pathways. This topic could be better addressed in the Discussion.

Yes, most people think that vitamin D deficiency should not prevail in countries with abundant sunshine like Kuwait. There is also strong believe that laboratory measurements are erroneous and

tend to underestimate vitamin D (there is some literature that support this point). With respect to the cutoff point, vitamin D deficiency is a major health problem no matter what is the cutoff point we use in our setting (i.e. More than one third of our adolescents had severe vitamin D deficiency: 25-OH-D levels < 25.0 nmol/L (10 ng/mL).

The best way to address underlying causes of this believe is to conduct a qualitative research. As a cross-section study this study aimed to confirm the existence and measure the size of the problem so that doubts about the problem should recede.

- In the sentence: “An optimal vitamin D level is essential during adolescents” – Please replace “adolescents” by “adolescence”.

Thank you. This is corrected now.

STROBE checklist

- Items #3, 4, 8, 9, 10, 12C and 16A were not filled in the checklist.

We have done this now.

Reviewer: 2

Comments to the authors

General Comments

This is a descriptive study that investigated prevalence and determinants of vitamin D deficiency in adolescents in Kuwait. Whilst vitamin D status and predictors of deficiency are well established in some areas of the world and in the adult population, there is still need for more studies on this topic in the adolescent group and in areas of the world such as the Middle East. Adolescence is a critical period for growth and development affected also by changes in behavior that might affect status of vitamin D. Thus this paper extends beyond local interest. The paper would benefit from a linguistic review.

Thank you. We have edited the manuscript now.

Title: Relevant and includes study design.

Abstract: Important information is missing from the abstract especially in relation to the methods of the study. For example in the methods section, the authors do not include essential information on participants (e.g. characteristics such as age spectrum, participation rate) and on outcome variables i.e. what data was collected from parents and children and what tools were used to do that? In the results section prevalence should be reported as a percentage and the number of participants in the brackets. In the conclusion, do not include statements such as “which may reflect sun avoidance” unless can support this from your findings.

We added now the age range, methods of data collection from parents and adolescents (See abstract). The outcome variable is vitamin D status and this is in the abstract. Because prevalence of vitamin D is >80%, lack of exposure to sunlight cannot be overlooked as potential explanation. We have discussed the sun exposure in the discussion in details and thus cannot ignore this point in the abstract.

Strengths and Limitations

This section should not include results thus second bullet point is not relevant. Strengths of the study is its study design since cross-sectional studies are appropriate to measure prevalence. Was the participation rate high? If yes, this is also a strength.

We agree with the comment and have deleted the results from strengths and limitation. All details of

the response% is described in the results section (please see firsts paragraph in the results section, page 8). "Of 1583 parents approached, 161 refused to participate (children, parents or both). Another 6 samples were not sufficient to conduct blood analysis, thus the analysis below comprised 1416 students"

Introduction: This section is very brief. Whilst the first paragraph sets the scene as to the importance of investigating vitamin D status in adolescents, the rest of this section fails to describe what is already known and what this study will add. For example the authors claim that other studies that investigated vitamin D status (especially in the Middle East area) suffer from methodological problems. What are these problems and how has this current study dealt with these problems in order to provide further evidence on the topic of interest? Also the authors failed to explain how this study might be of international interest.

We have now added the weaknesses of the previous studies (See Introduction, Page 5).

Methodology: The methods section is informative and includes most of the items as per STROBE. Statistical analysis is appropriate and valid. However it will benefit from sub-sections to make it more structured.

We agree with the comment. We added subsection (subtitles for The Methods and Results sections). Please see The Methods and The Results.

Information not included but important is:

1. Overall number of adolescents in the age group 11-16 in Kuwait and number of adolescents in the selected sample size i.e. there are x number of adolescents aged 11-16 in all governorates of Kuwait, the overall sample of adolescents attending the 12 selected middle public schools were 1583.

This is available in the result section. Please see first paragraph in The Result section (page 8).

2. Definition and measurement of variables: the authors provide information on how some variables were measured and their definition as per tools used in other studies. For the rest of the variables definition can be found in the tables. It would be good to include in the body text under methods (e.g. education was measured as no education- primary/intermediate –secondary – diploma- university). We agree with the comment and added this now to the methods (See The Methods section, page 6).

3. Season/period of measurements – seasonality is an important determinant of vitamin D status and information on the period of measurement is found in later sections of the paper and not in methods

We mentioned this point in the discussion to explain why we couldn't explore seasonality. If we also mentioned this in the methods section, this will be deemed as a repetition.

4. Setting/location: This is not clear and does not allow replication of study methods

We are not sure what does this comment mean. Our setting/location is the state of Kuwait. We specified the latitude (29.3759° N) of the country.

5. Reference to patient involvement in the design of the study is not made (as per the journal guidelines)

We have added this to The Methods Section now. See page 8.

Results: The results section provides a good description of the study results including information on

participants (numbers and characteristics) and outcome data reported as population prevalence per vitamin D category (as described in methods). Also authors refer to subsidiary analysis very effectively showing that their data analysis was comprehensive. Tables and figures are clear. Sub sections would provide better structure to this section.

As mentioned above we agree with the comment and added now subsections.

Also in regards to this section please note the following:

1. First paragraph: apart from gender and mean age please include other characteristics of participants e.g. ethnicity, parental education and household income)

There is no clear and well-recognized classification of ethnicity in Kuwait. Parental education and income are in Table 1.

2. How can the median vitamin D level be 29 nmol/L but 80% of participants have a vitamin D level < 20 nmol/L. How are vitamin D levels distributed in the participants? It might be good inserting a figure that shows the distribution of vitamin D levels amongst participants as a continuous variable.

It is not written anywhere in the manuscript that "80% of participants have a vitamin D level < 20 nmol/L". If you mean that 81.21% of the participants are vitamin D deficient, then see the definition of vitamin D deficiency, which is <50 nmol/L not 20 nmol/L. This is presented in the method section and the results section. The information about the distribution of vitamin D as a continuous variable can be gleaned from Figure 1. In The Result section we showed that the median (Interquartile Range, IQR) of 25-OH-D was 29.7 (19.2- 44.2) nmol/L. This is a numerical presentation of the distribution of vitamin D.

3. As indicated in the abstract, please report prevalence as % and include (n) in the brackets We used brackets not only for %s but also for the 95% CIs. We also looked at the current publications on the journal website and did not find this style.

Discussion:

This section requires more structure and a more focused interpretation of the results of the study. Once more sub sections will help. The first paragraph should include a summary of results. The authors attempted to start this section with a summary of results but only referred to vitamin D status and not determinants.

If we list the factors associated with vitamin D deficiency in the summary at the beginning of the discussion, it will be just repetition of what we said at the end of The Results Section (just in the previous page). Authors are not supposed to repeat the result in the discussion.

Discussion on how vitamin d status compared with other countries is satisfactory.

Thank you.

Discussion in regards to the determinants of vitamin D deficiency in Kuwait is discussed next. Generally authors have discussed and interpreted those findings in the context of other studies and the strengths and limitations of their own study. However the discussion on sun exposure is not structured well. Authors start to discuss the skepticism amongst Arab states in regards to the validity of the results which is not part of the aims of the study. They also start to discuss determinants of low sun exposure referring to variables they have not measured such as dust storms and skin type. They need to emphasize first which of the parameters they have measured show that exposure to sun is

low and whether these parameters are linked to vitamin D deficiency in their own population (e.g. time spent out in the sun, times walking to school etc.). Authors can then refer to other parameters relating to sun exposure that they have not measured (e.g. skin type, dust storms) and how this might have also contributed to low vitamin D.

Cross-sectional studies are used to measure the size of health problems. Hence the ultimate goal of our study was to reduce skepticism about the existence and the size of the problem. We have made this point clearer now in the discussion (See The Discussion, Fourth Paragraph, Page 14). We also now highlighted the findings related to the exposure to sunlight as you have suggested (See page 14).

Conclusion: this section should provide recommendations for the future based on the results of the study. Reference to skepticism and cynicism is not relevant. Instead authors should propose ways on how policy makers can use these results to improve vitamin D status in adolescents in Kuwait.

As mentioned above the goal of our study (like any other cross-sectional study) is to clear doubts about the existence and the size of the problem. This is a prerequisite to take action (i.e. size of the problem is an important point in setting priorities for prevention). We have modified the conclusion to show this point (See page 17-18). We proposed encouragement adequate outdoor daytime activities, Vitamin D fortification particularly for food products that are popular for adolescents and vitamin D supplementation. All these are relevant to the findings of the study and they are covered in the discussion.

VERSION 2 – REVIEW

REVIEWER	Kelly Virecoulon Giudici Postdoctoral researcher at the Nutrition Department, School of Public Health, University of São Paulo, Brazil
REVIEW RETURNED	15-Mar-2018

GENERAL COMMENTS	English was revised and the manuscript quality has improved. However, minor changes are still needed, since there are some comments that were not satisfactorily answered or corrected in the manuscript. Results  • Previous comment: In Table 1, I suggest including two columns (male and female), in addition to the total column. It would also be valuable to include 25(OH)D values (in all columns: total, male and female). Authors response: Table 1 meant to present socio-demographic data of the study group. Including vitamin D level in the table will make the title of table inappropriate (i.e. vitamin D is not a socio-demographic characteristic). The information requested in the comment above are all available in the text. This include vitamin D level among males, females and in the total study group. New comment: In Table 1, I suggested including two columns (male and female) to verify if there were differences in all sociodemographic characteristics according to sex. This information still cannot be found in the text, as said by the authors in their response letter.  • Previous comment: In the text: “Prevalence of vitamin D deficiency was 1,150 (81.21%, 95%CI: 71.61-90.81) while severe deficiency was 559 (39.48%). Only 51 (3.60%) were vitamin D sufficient.
--

Prevalence was significantly higher among females compared to males (91.69% vs. 70.32%; $p < 0.001$).” – Which prevalence does the second sentence refer to: vitamin D deficiency, severe deficiency or sufficiency? Analyzing the percentage, one can presume that it refers to vitamin D deficiency. However, the sentence is not clearly written.

Authors response: We have modified this to become clearer. I.e. vitamin D deficiency not severe deficiency (see first paragraph, page 10)

New comment: This question was not corrected in the text: “Prevalence of vitamin D was significantly higher among females compared to males (91.69% vs. 70.32%; $p < 0.001$)”. The new sentence still does not enable to understand if authors refer to vitamin D deficiency, severe deficiency or sufficiency. Please complete the sentence.

- Previous comment: Please rewrite the following sentence to avoid redundancy: “Factors that showed significant association with vitamin D deficiency (25-OH-D < 50 nmol/L) in univariate analysis were Gender, parental education, number of times per week the participants consumed breakfast and dinner prepared outside home, taking supplements, walking to school (instead of using school bus or car) in addition to BMI were all significant predictors in univariate analysis (Supplementary Table).”

Authors response: It seems that you mean the data is presented in the text and the supplementary table. As such, we have shorten this paragraph to one sentence referring to the supplementary table. Because we cannot keep a paragraph with only one sentence, we merged this sentence with the next paragraph. (see The Results section, first paragraph, page 12)

New comment: No, I was referring to the redundancy inside the sentence, given that it started by “Factors that showed significant association with vitamin D deficiency (25-OH-D < 50 nmol/L) in univariate analysis were...” and ended with “...were all significant predictors in univariate analysis” (the term “univariate analysis” is doubled). The information should be kept in the text. The redundancy could have been avoided by changing the sentence for the following two options:

1. “Factors that showed significant association with vitamin D deficiency (25-OH-D < 50 nmol/L) in univariate analysis were gender, parental education, number of times per week the participants consumed breakfast and dinner prepared outside home, taking supplements, walking to school (instead of using school bus or car) in addition to BMI (Supplementary Table).”

2. “Gender, parental education, number of times per week the participants consumed breakfast and dinner prepared outside home, taking supplements, walking to school (instead of using school bus or car) in addition to BMI were all significant predictors of vitamin D deficiency (25-OH-D < 50 nmol/L) in univariate analysis (Supplementary Table).”

- Previous comment: Tables 1, 2 and supplementary table: Instead of stating the number of missings in the footnote (“1Missing for 33 participants; 2 Missing for 20 participants; 3 Missing for 46 participants; 4Missing for 28 participants; 5Missing for 19 participants”), I suggest including the real “n” for each variable with parenthesis right after the variable’s name Ex: “Father’s education (n = 1383)”.

Authors response: It would be much easier for readers to assess the impact of missing values if they are mentioned this way. As you can

see the number of missing are relatively small compared to our sample and thus the message is that missing are unlikely to change our findings significantly. Presenting real numbers instead of the number of missing will imply that readers have to calculate the number of missing values themselves.

New comment: I do not agree. As a reader, it was not easy to verify the real n for each variable with the missings given at the footnote (with superscript numbers identifying each variable). In this case, the readers have to calculate the number of real n for each variable themselves. Thus, I suggest verifying and adopting the model that is more usual for this journal, according to previous publications.

- Previous comment: Also, the number of missings seem to be incorrect in some variables. Example: in table 2, title says “1,416 middle school students”. However, there are 488 boys and 662 girls ($488 + 662 = 1,150$) and no missing information is given for gender. For “Governorate”, the sum is $127+188+183+214+124+296=1,132$. Adding the number of missings ($1,132 + 20$), we get 1,152, and not 1,416 as said in the title. This fact also happens with other variables, so please revise all numerical information and provide the correct information in all tables.

Authors response: There are no mistakes here. The title of the column where these numbers existed is “prevalence n (%)” hence the next column is prevalence ratio. Because this is a cross-sectional study, the typical data presentation is to show the prevalence of the outcome among each exposure category. There are 488 boys with vitamin D deficiency out of the total number of boys which is 694. Similarly, there were 662 girls with vitamin D deficiency out of 722 girls. In order to remove any confusion, we made the title of the column clearer now saying that this is the “prevalence of vitamin D deficiency”. (See supplementary table and Table 2)

New comment: Thank you for the explanation and for changing the title of the column. The table is now easier to understand, but I suggest including the total n of vitamin D deficiency on the top of the column “prevalence of vitamin D deficiency” ($n = 1,150$). For the variable “Governorate”, the sum is $127+188+183+214+124+296=1,132$. Adding the number of missings ($1,132 + 20$), we get 1,152, and not 1,150 as it should be. If this missing value in the footnote corresponds to the total sample (20 from 1,416), it should not. The reported missings in this table should be according to the total n of the column ($n=1,150$). Another example: for the variable “Consumption of sugary drinks per week” the sum is $123+633+135+256=1,147$, and there is no missing information in the footnote (once again, the n is different from 1,150). So, I reinforce the need for carefully revising all numerical information.

- Previous comment: The percentages described for all variables in Table 2 and supplementary table 1 are visibly incorrect. Example for BMI categories: if 465 subjects are normal weight, 270 are overweight, 400 are obese and 15 are underweight, the percentages should be, respectively, 40.4%, 23.5%, 34.8% and 1.3% (summing 100%), and not 77.37%, 84.38%, 84.93% and 62.50% as given. Again, the total n for this variable ($n=465+270+400+15=1,150$), considering additional 26 missings ($1,150+26=1,176$), is different from the title ($n=1,416$).

Authors response: Please see the response to the comment above. These are not the numbers and %s of participants who are normal weight, overweight or obese. These are the numbers and %s of

	participants who had vitamin D deficiency in normal weight, overweigh and obese. i.e. prevalence of the outcome in each exposure category. Tables can only be understood properly if the column titles and row titles are taken into account. New comment: Thank you for the explanation. As the authors have just said, "tables can only be understood properly if the column titles and row titles are taken into account". This was not easy to understand in the first version of the manuscript, when the column did not have a title.  • Previous comment: Figure 1 should present the correlation line and the correlation coefficient between vitamin D and PTH. In addition, please add the unit of measure for 25(OH)D (horizontal axis). Authors response: We have added the regression line to the figure now. We have added the unit to the X-axis. (See figure 1) New comment: Adding the regression coefficient and the p-value (in the corner) would be valuable to the figure.
--	--

REVIEWER	Ourania Kolokotroni University of Nicosia
REVIEW RETURNED	02-Apr-2018

GENERAL COMMENTS	The authors did not respond to a number of the comments. The paper still requires linguistic review prior to publication
--

VERSION 2 – AUTHOR RESPONSE

Reviewer: 1

English was revised and the manuscript quality has improved. However, minor changes are still needed, since there are some comments that were not satisfactorily answered or corrected in the manuscript.

Thank you. Below is the description for the changes we made based on your new comments.

Results

- Previous comment: In Table 1, I suggest including two columns (male and female), in addition to the total column. It would also be valuable to include 25(OH)D values (in all columns: total, male and female).

Authors' response: Table 1 meant to present socio-demographic data of the study group. Including vitamin D level in the table will make the title of table inappropriate (i.e. vitamin D is not a socio-demographic characteristic). The information requested in the comment above are all available in the text. This include vitamin D level among males, females and in the total study group.

New comment: In Table 1, I suggested including two columns (male and female) to verify if there were differences in all socio-demographic characteristics according to sex. This information still cannot be found in the text, as said by the authors in their response letter.

Response: Table 1 has been modified as suggested. Please note that we had to modify the title of the table so that it reflects the content. All %s are column percentages add up to 100% and the number add up to the total except when there is missing values. Missing values are no longer in footnotes

rather we put “N” beside each variable inside the table as suggested. The differences in socio-demographic factors between males and females have been adjusted for in multivariate logistic regression.

- Previous comment: In the text: “Prevalence of vitamin D deficiency was 1,150 (81.21%, 95%CI: 71.61-90.81) while severe deficiency was 559 (39.48%). Only 51 (3.60%) were vitamin D sufficient. Prevalence was significantly higher among females compared to males (91.69% vs. 70.32%; $p < 0.001$).” – Which prevalence does the second sentence refer to: vitamin D deficiency, severe deficiency or sufficiency? Analyzing the percentage, one can presume that it refers to vitamin D deficiency. However, the sentence is not clearly written.

Authors response: We have modified this to become clearer. I.e. vitamin D deficiency not severe deficiency (see first paragraph, page 10)

New comment: This question was not corrected in the text: “Prevalence of vitamin D was significantly higher among females compared to males (91.69% vs. 70.32%; $p < 0.001$)”. The new sentence still does not enable to understand if authors refer to vitamin D deficiency, severe deficiency or sufficiency. Please complete the sentence.

Response: This is corrected now (See page 11, third line)

- Previous comment: Please rewrite the following sentence to avoid redundancy: “Factors that showed significant association with vitamin D deficiency (25-OH-D < 50 nmol/L) in univariate analysis were Gender, parental education, number of times per week the participants consumed breakfast and dinner prepared outside home, taking supplements, walking to school (instead of using school bus or car) in addition to BMI were all significant predictors in univariate analysis (Supplementary Table).”

Authors response: It seems that you mean the data is presented in the text and the supplementary table. As such, we have shortened this paragraph to one sentence referring to the supplementary table. Because we cannot keep a paragraph with only one sentence, we merged this sentence with the next paragraph. (see The Results section, first paragraph, page 12)

New comment: No, I was referring to the redundancy inside the sentence, given that it started by “Factors that showed significant association with vitamin D deficiency (25-OH-D < 50 nmol/L) in univariate analysis were...” and ended with “...were all significant predictors in univariate analysis” (the term “univariate analysis” is doubled). The information should be kept in the text. The redundancy could have been avoided by changing the sentence for the following two options:

1. “Factors that showed significant association with vitamin D deficiency (25-OH-D < 50 nmol/L) in univariate analysis were gender, parental education, number of times per week the participants consumed breakfast and dinner prepared outside home, taking supplements, walking to school (instead of using school bus or car) in addition to BMI (Supplementary Table).”
2. “Gender, parental education, number of times per week the participants consumed breakfast and dinner prepared outside home, taking supplements, walking to school (instead of using school bus or car) in addition to BMI were all significant predictors of vitamin D deficiency (25-OH-D < 50 nmol/L) in univariate analysis (Supplementary Table).”

Response: We have brought this paragraph back and made the changes as per suggestion. See page 13.

- Previous comment: Tables 1, 2 and supplementary table: Instead of stating the number of missings in the footnote (“1Missing for 33 participants; 2 Missing for 20 participants; 3 Missing for 46 participants; 4Missing for 28 participants; 5Missing for 19 participants”), I suggest including the real “n” for each variable with parenthesis right after the variable’s name Ex: “Father’s education (n = 1383)”.

Authors response: It would be much easier for readers to assess the impact of missing values if they are mentioned this way. As you can see the number of missing are relatively small compared to our sample and thus the message is that missing are unlikely to change our findings significantly. Presenting real numbers instead of the number of missing will imply that readers have to calculate the number of missing values themselves.

New comment: I do not agree. As a reader, it was not easy to verify the real n for each variable with the missings given at the footnote (with superscript numbers identifying each variable). In this case, the readers have to calculate the number of real n for each variable themselves. Thus, I suggest verifying and adopting the model that is more usual for this journal, according to previous publications.

Response: We have now followed your suggestion in all tables. Please see all tables.

- Previous comment: Also, the number of missings seem to be incorrect in some variables. Example: in table 2, title says "1,416 middle school students". However, there are 488 boys and 662 girls ($488 + 662 = 1,150$) and no missing information is given for gender. For "Governorate", the sum is $127+188+183+214+124+296=1,132$. Adding the number of missings ($1,132 + 20$), we get 1,152, and not 1,416 as said in the title. This fact also happens with other variables, so please revise all numerical information and provide the correct information in all tables.

Authors response: There are no mistakes here. The title of the column where these numbers existed is "prevalence n (%)" hence the next column is prevalence ratio. Because this is a cross-sectional study, the typical data presentation is to show the prevalence of the outcome among each exposure category. There are 488 boys with vitamin D deficiency out of the total number of boys which is 694. Similarly, there were 662 girls with vitamin D deficiency out of 722 girls. In order to remove any confusion, we made the title of the column clearer now saying that this is the "prevalence of vitamin D deficiency". (See supplementary table and Table 2)

New comment: Thank you for the explanation and for changing the title of the column. The table is now easier to understand, but I suggest including the total n of vitamin D deficiency on the top of the column "prevalence of vitamin D deficiency" ($n = 1,150$). For the variable "Governorate", the sum is $127+188+183+214+124+296=1,132$. Adding the number of missings ($1,132 + 20$), we get 1,152, and not 1,150 as it should be. If this missing value in the footnote corresponds to the total sample (20 from 1,416), it should not. The reported missings in this table should be according to the total n of the column ($n=1,150$). Another example: for the variable "Consumption of sugary drinks per week" the sum is $123+633+135+256=1,147$, and there is no missing information in the footnote (once again, the n is different from 1,150). So, I reinforce the need for carefully revising all numerical information.

Response: In order to sort this problem out, we have made a new column that presents the "n" in each category in each variable; and now all numbers can be reproduced including the prevalence in each group and even the prevalence ratios in the supplementary Table. We hope this will remove any confusion. We also have made the change you requested (i.e. reporting $n=1150$ in the column). We also removed footnotes about missing values and reported the actual numbers beside each category of all variables.

- Previous comment: The percentages described for all variables in Table 2 and supplementary table 1 are visibly incorrect. Example for BMI categories: if 465 subjects are normal weight, 270 are overweight, 400 are obese and 15 are underweight, the percentages should be, respectively, 40.4%, 23.5%, 34.8% and 1.3% (summing 100%), and not 77.37%, 84.38%, 84.93% and 62.50% as given. Again, the total n for this variable ($n=465+270+400+15=1,150$), considering additional 26 missings ($1,150+26=1,176$), is different from the title ($n=1,416$).

Authors response: Please see the response to the comment above. These are not the numbers and %s of participants who are normal weight, overweight or obese. These are the numbers and %s of participants who had vitamin D deficiency in normal weight, overweigh and obese. i.e. prevalence of the outcome in each exposure category. Tables can only be understood properly if the column titles and row titles are taken into account.

New comment: Thank you for the explanation. As the authors have just said, “tables can only be understood properly if the column titles and row titles are taken into account”. This was not easy to understand in the first version of the manuscript, when the column did not have a title.

Response: Thank you. The changes we made now will hopefully remove any confusion.

- Previous comment: Figure 1 should present the correlation line and the correlation coefficient between vitamin D and PTH. In addition, please add the unit of measure for 25(OH)D (horizontal axis).

Authors response: We have added the regression line to the figure now. We have added the unit to the X-axis. (See figure 1)

New comment: Adding the regression coefficient and the p-value (in the corner) would be valuable to the figure.

Response: We have added now the correlation coefficient and p-value at the upper right corner of the figure.

Reviewer: 2

Reviewer Name: Ourania Kolokotroni

Institution and Country: University of Nicosia

Please state any competing interests or state ‘None declared’: None

Please leave your comments for the authors below The authors did not respond to a number of the comments. The paper still requires linguistic review prior to publication

Response: The manuscript has been thoroughly reviewed and edited now; and further corrections can be done during copy editing.

VERSION 3 – REVIEW

REVIEWER	Kelly Virecoulon Giudici Postdoctoral researcher at the Nutrition Department, School of Public Health, University of São Paulo, Brazil
REVIEW RETURNED	14-May-2018

GENERAL COMMENTS	Most changes were made satisfactorily and the manuscript is in better shape. A minor suggestion can be found below. Previous comment 1: In Table 1, I suggest including two columns (male and female), in addition to the total column. It would also be valuable to include 25(OH)D values (in all columns: total, male and female). Authors response 1: Table 1 meant to present socio-demographic data of the study group. Including vitamin D level in the table will make the title of table inappropriate (i.e. vitamin D is not a socio-demographic characteristic). The information requested in the
---

	comment above are all available in the text. This include vitamin D level among males, females and in the total study group. Previous comment 2: In Table 1, I suggested including two columns (male and female) to verify if there were differences in all sociodemographic characteristics according to sex. This information still cannot be found in the text, as said by the authors in their response letter. Authors' response 2: Table 1 has been modified as suggested. Please note that we had to modify the title of the table so that it reflects the content. All %s are column percentages add up to 100% and the number add up to the total except when there is missing values. Missing values are no longer in footnotes rather we put "N" beside each variable inside the table as suggested. The differences in socio-demographic factors between males and females have been adjusted for in multivariate logistic regression. >>>> New comment: It seems that authors did not correctly understand my previous comment. Usually when presenting descriptive data according to sex, tables inform when any variable presents statistical difference (given in this case, by Pearson's Chi-square test for the categorical variables, and Student's T-test for the continuous variable – vitamin D). For the variables with differences, a simple "*" at the end of the line with "**P<0.05" at the footnote would be enough.
--	---

REVIEWER	Ourania Kolokotroni University Of Nicosia Medical School
REVIEW RETURNED	21-May-2018

GENERAL COMMENTS	The manuscript still requires linguistic review. Also a number of the comments either minor or major have not been addressed by the authors for both reviewers. One example of the comment on providing the season (period) of data collection. This is common practice to report period of data collection even when seasonality is not an important factor- even more so in such a study when seasonality is important. The authors were asked to include this in the methods section and they have not done so. Another example is reporting prevalence as crude numbers. this is not right but authors insist on this. However my main concerns are with regards to the short introduction which fails to set the scene and describe existing knowledge and need for the current study. When referring to existing knowledge, the authors have edited introduction as per some comments but not completely. For example, they have added a comment in regards to existing knowledge from clinical trials but have (a) not provided what that is and (b) not referenced it. I would be happy to accept this paper for publication if authors address completely all the reviewers' comments
--

VERSION 3 – AUTHOR RESPONSE

Reviewer #1:

All comments by this Reviewer have been addressed in the manuscript.

Reviewer #2:

Comment: The manuscript still requires linguistic review.

Answer: We wish if The Reviewer pointed out to one of two sentences in which there are grammatical errors or ambiguity. We have gone through the manuscripts several times and made a few amendments. If you think our manuscript should go to editing services, we are happy to do so upon your request.

Comment: Also a number of the comments either minor or major have not been addressed by the authors for both reviewers.

Answer: We do not know which comment(s) The Reviewer is talking about. We do not also know which one is minor and which one is major. Her general comment above cannot be justified because we addressed every point she raised in the first response to her review. That means we either followed what she suggested or we explained why we preferred to keep things as they are. Please see below our response to the examples she wrote below.

Comment: One example of the comment on providing the season (period) of data collection. This is common practice to report period of data collection even when seasonality is not an important factor- even more so in such a study when seasonality is important. The authors were asked to include this in the methods section and they have not done so.

Our previous response to this comment was “We mentioned this point in the discussion to explain why we couldn’t explore seasonality. If we also mentioned this in the methods section, this will be deemed as a repetition.”

Thus, we addressed this point from the first time by referring to the information she requested in the discussion. Her comment above implies that the information is not there at all in the paper! Overall, is this a minor or a major issue!

Now, based on her new comment above, we added the data collection period to the methods section (see page 6) in addition to the previously written statement in discussion, page 15.

Comment: Another example is reporting prevalence as crude numbers. this is not right but authors insist on this.

Her previous comments and our response were as follows: her comments: In the results section prevalence should be reported as a percentage and the number of participants in the brackets. & As indicated in the abstract, please report prevalence as % and include (n) in the brackets. Our response: We used brackets not only for %s but also for the 95%CIs. We also looked at the current publications on the journal website and did not find this style.)

Thus, we reported the prevalence as a percentage and the 95%CI of the percentage in order to show how precise our estimate. This also means, we addressed this point in the first review. If our response was not satisfactory, she could have told us in the second review!

Therefore, we did not report prevalence as crude number rather as n (%; 95%CI: ##- ## %) a style that was found in many BMJ articles. Using the same style our abstract was accepted in two major meetings (Society for Epidemiologic Research (SER) 51st Annual Meeting to be held in Baltimore, Maryland, USA; June 20-22, 2018 & 2018 Society for Pediatric and Epidemiologic Research (SPER) Annual Meeting, to be held in Baltimore, Maryland, USA; June 18-19, 2018).

Now in order to satisfy her comment, we removed the absolute value numbers and left only %. The style presented now is identical to the style in the BMJ open journal (see a recent example below from the same journal). We also think this is a minor issue. We are also happy that you change the style to any form you think appropriate.

Reference:

Xiao W, Chen X, Yan W, Zhu Z, He M. Prevalence and risk factors of epiretinal membranes: a systematic review and meta-analysis of population-based studies. *BMJ open* 2017;7(9):e014644 doi: 10.1136/bmjopen-2016-014644.

Comment: However my main concerns are with regards to the short introduction which fails to set the scene and describe existing knowledge and need for the current study. When referring to existing knowledge, the authors have edited introduction as per some comments but not completely. For example, they have added a comment in regards to existing knowledge from clinical trials but have (a) not provided what that is and (b) not referenced it.

This comment should reassure everyone because it implies that there is no concern on The Methodology, The Results or The Discussion. Scientists should worry most when there is a problem in the methodology, results or discussion. In the introduction, as long as the objectives are clearly stated, everything else can be considered a minor issue. Nevertheless, we thank The Reviewer for the specific point she raised above and have fully addressed the point through citing recent systematic reviews of RCTs for each specific outcome including cancer, asthma, cardiovascular diseases and overall mortality. See first paragraph in the introduction.

Comment: I would be happy to accept this paper for publication if authors address completely all the reviewers' comments

We truly believe that the ultimate decision to “accept” or “reject” must be in the hand of The Editor not The Reviewers. Reviewers can only recommend or provide their professional opinion. Therefore, a statement or a comment like the above should not be written by any reviewer.